# Wavelet Analysis on Groundwater, Surface-Water Levels and Water Temperature in Doñana National Park (Coastal Aquifer in Southwestern Spain)

Jennifer Treviño [1], Miguel Rodríguez-Rodríguez [2], María José Montes-Vega [2], Héctor Aguilera [3], Ana Fernández-Ayuso [4] and Nuria Fernández-Naranjo [5,*]

1. Geology and Geophysics Department, Texas A&M University, College Station, TX 77843, USA
2. Department of Physical, Chemical and Natural Systems, Pablo de Olavide University, Utrera Road, Km 1., 41013 Seville, Spain
3. Spanish Geological Survey-CSIC, Rios Rosas, 23, 28003 Madrid, Spain
4. Faculty of Sciences, Autónoma University of Madrid, Campus of Cantoblanco, C/Francisco Tomás y Valiente 7, 28049 Madrid, Spain
5. Centre of Hydrogeology, University of Málaga (CEHIUMA), 29071 Málaga, Spain
* Correspondence: nuria.naranjo.phd@gmail.com

**Abstract:** The Doñana National Park (DNP) is a protected area with water resources drastically diminishing due to the unsustainable extraction of groundwater for agricultural irrigation and human consumption of a nearby coastal city. In this study, we explore the potential of wavelet analysis applied to high-temporal-resolution groundwater-and-surface-water time series of temporary coastal ponds in the DNP. Wavelet analysis was used to measure the frequency of changes in water levels and water temperature, both crucial to our understanding of complex hydrodynamic patterns. Results show that the temporary ponds are groundwater-dependent ecosystems of a through-flow type and are still connected to the sand-dune aquifer, regardless of their hydrological affection, due to groundwater withdrawal. These ponds, even those most affected by pumping in nearby drills, are not perched over the saturated zone. This was proven by the evidence of a semi-diurnal (i.e., 6 h) signal in the surface-level time series of the shallow temporary ponds. This signal is, at the same time, related to the influence of the tides affecting the coastal sand-dune aquifer. Finally, we detected other hydrological processes that affect the ponds, such as evaporation and evapotranspiration, with a clear diurnal (12 h) signal. The maintenance of the ecological values and services to the society of this emblematic wetland is currently in jeopardy, due to the effect of the groundwater abstraction for irrigation. The results of this study contribute to the understanding of the behavior of these fragile ecosystems of DNP, and will also contribute to sound-integrated water-resource management.

**Keywords:** surface-water–groundwater interactions; groundwater withdrawal; tidal effect; protected area

## 1. Introduction

Established as a UNESCO World Heritage Site in 1994, Doñana National Park is a protected area with a rich diversity of biota, flora, and fauna. However, scientists have recently sounded the alarm on the endangerment of water resources in this area due to the illegal extraction of water for agricultural purposes and the recent legalization of unregulated groundwater pumping, which has further exacerbated the problem [1,2]. It must be pointed out that the situation has become alarming recently. Due to the last intense 8-year drought and high evaporation rates, even the Santa Olalla (SOL) pond was reported as being completely dried out in September 2022. The SOL pond was reported previously to have been desiccated only twice: in 1983 and again in 1995 [3]. As in the majority of Mediterranean wetlands, the availability of shallow water bodies such as ponds, which

are essential for waterfowl breeding, strongly depends on groundwater discharge from the aquifer. All efforts made for a better understanding of the hydrogeological functioning of the aquifer and related groundwater-dependent ecosystems should be considered. Knowledge in this case is crucial to significantly improve the water-management measures to overcome the many risks are encompassed in the ecosystems of Doñana. Despite the significance of groundwater-dependent ecosystems for the maintenance of the ecological processes in Doñana, research into the hydrological processes of this vast wetland area started merely two decades ago. The Doñana ponds were typified according to their hydrology and chemical composition [4,5], their hydrology [6] and their hydrochemistry [7]. Most recently, a Spanish monography on Doñana's ponds was published, [8] focused on the ecological characteristics of the most relevant ponds. The bigger size and longer hydroperiod of the Santa Olalla and Dulce ponds made them suitable for many limnological [9–12] and hydrogeological studies [13–15]. Surface-water–groundwater interaction has also been studied under different perspectives, such as vegetation–groundwater relations [16], climate change [17] and landscape management [18].

Over the course of one year (May 2021–May 2022), we examined four of the most representative ponds within the DNP and analyzed their hydrological functioning and relation with the aquifer. The main method used to interpret data collected in the field was through wavelet analysis [19,20], a statistical technique that has been increasingly used in a hydrogeologic context to help better understand groundwater and surface-water interactions under non-stationary assumptions [15,21]. This is because wavelet transforms can be used to discern minute changes presented throughout time, indicating the variability and complexity of aquifer systems. Wavelet analysis used in the field of hydrology has been applied since the 1990s for the multi-temporal scale analysis of hydrometeorological series, with increasing application on this method in the last decade [21–23]. Although most studies focus on the analysis of long-climatic and -hydrological time series to reveal inter-annual components [23,24], it has been shown [15] that this methodology is also helpful for shorter time series with a higher sampling frequency (e.g., hourly). The interpretation of the hydrological processes in a given water body (e.g., river, pond, playa-lake, aquifer, etc.), derived from a high-temporal resolution but short (e.g., annual) time series will be different from a low-temporal resolution but long time series (e.g., decadal). For this particular study, we have used a high-resolution (3 h) yearlong time series to interpret the hydrological behavior of a series of ponds. The main reasons for this approach are: Firstly, the authors carried out a previous study in the same area, but focused on the wavelet analysis of piezometric levels during a 2 year timespan [15] to enhance the hydrogeological conceptual model in one particular permanent pond, not affected by anthropic disturbances (SOL pond). However, here, the focus has been extended to four ponds, some of them affected by anthropic disturbances [14], where surface and groundwater level and temperature time series have been compared. Secondly, for the first time, one of the ponds studied (DUL pond) has been monitored with this high-temporal resolution. Finally, the general hydrogeological behavior of the unconfined sand-dune aquifer has been previously described, but not the detailed interactions of this aquifer with some of the sand-dune ponds included in this paper. The leading objectives of the following study are to (i) interpret the high-frequency hydrologic function of the most representative ponds of DNP, (ii) better understand groundwater and surface-water interactions in these dependent ecosystems, and (iii) assess possible hydrologic impact on coastal ponds due to nearby groundwater extraction.

## 2. Study Site

The DNP is a UNESCO World Heritage Site in southwestern Spain (37° N, 6° W) at the core of a huge wetland placed at the end of the Guadalquivir River delta. The Park has an extension of 54,252 ha, but it is connected to the hydrological catchment, which is much bigger and covers an area of 260,000 ha. Originally, the area was a seasonal-marsh system geologically formed by the siltation of a big estuary, approximately over the last

1000 years. The estuary was slowly filled with silts and clays and transformed into a dynamic river, channel, and levee system. Then, a tidal marsh was formed and, finally, a fluvial-pluvial marsh filled the original estuary. The original marsh surface area has been transformed since 1920 and, at the present time, only one fifth of the surface remains in the original natural conditions [25]. The climate is sub-humid and typically has rainfall of 550 mm per year, normally occurring between October and April. Some changes in temporal distribution in the rainfall days have been found by Naranjo-Fernandez et al. [26], highlighting an increase in torrentiality. It rains with the same amount of water, but this is distributed on fewer days per year. The marsh is fed to a certain extent by a series of seasonal streams and, mainly, by direct precipitation. It is nowadays disconnected from the Guadalquivir River watercourse, although recent projects are taking into consideration its re-connection [27]. The marshes are dry during summer, due to high evapotranspiration rates. The marshes depend on rainfall and floods for their water supply, and are classified as a pulse system. Water levels and extent of inundation are determined by direct rainfall and, more importantly, by runoff arriving after rainfall in the river basin upstream, the Guadiamar River, an affluent of the Guadalquivir River. In the second half of the 20th century, the middle part of the Guadiamar basin and the marshes were anthropically disconnected by embankments. Consequently, the marshes became non-functional. Water was diverted to the Guadalquivir River.

A series of active and stabilized aeolian dunes separates the marsh from the coast. These dunes host a series of ephemeral and temporary ponds that are dependent on groundwater discharge. Nevertheless, the structure and geology of the aquifer are complex [28].

The Almonte–Marismas aquifer system is an aquifer containing mostly unconsolidated fine to medium-grained Plio-Quaternary materials deposited in alluvial, deltaic, aeolian, estuarine, and marshy environments. Such deposits overlie Pliocene and Miocene silt and marl [29], forming a single multi-layered series ranging from a few meters in the northern part (inland) to more than 200 m thick in the south (coastline). The aquifer has a total surface area of 2700 km$^2$, divided into a clayey area of around 1800 km$^2$—the marshland—and a sandy area of around 900 km$^2$, most of it placed on a 70 km coastal fringe going from the Guadalquivir River mouth to the Tinto River estuary and a dome-shaped area called Abalario. Both the sand fringe and the sand dome are called a "sand-dune aquifer" [30]. In the northern part of the aquifer, the sediments are mostly coastal, consisting of silts and sands. It is called the Deltaic Unit and goes from the Tinto River to the Guadiamar River (UD Unit in Figure 1. These Quaternary sediments were sequentially covered by sands and gravels when the deltaic sedimentary environment was replaced by rivers, channels, and levees, typical of an alluvial environment. Such coarse sediments do not outcrop in Doñana, because they were also gradually buried and replaced by different sedimentary rocks. This is called the Alluvial Unit, and it is depicted in CS-1 and in CS-2 of Figure 1 (UA Unit, see figure caption). The Alluvial Unit was first covered with fine-grained sediments, forming the so-called Marsh Unit: silts and clays that thicken from N (50 m) to S (70 m). See Figure 2 for more details (UM Unit). Secondly, the most recent sedimentary environment is called the Aeolian Unit. This Aeolian Unit covers the Marsh Unit in the SE part of the aquifer and the Deltaic Unit in the SW part (see Figure 2 and CS-1). This is a sandy-dune belt, partly active nowadays, that goes from the Tinto River estuary to the Guadalquivir River mouth. Finally, the Deltaic, Alluvial and Marsh Units are placed above the impervious substratum marls and silts of the so-called Miocene and Pliocene Units.

The Guadalquivir River Basin Authority (DHG) measures water levels in more than 300 piezometers across the aquifer. Some of the piezometric time series started in 1974. In many piezometers, groundwater levels are consistent with rainfall trends, but in other piezometers, levels have been found to be declining.

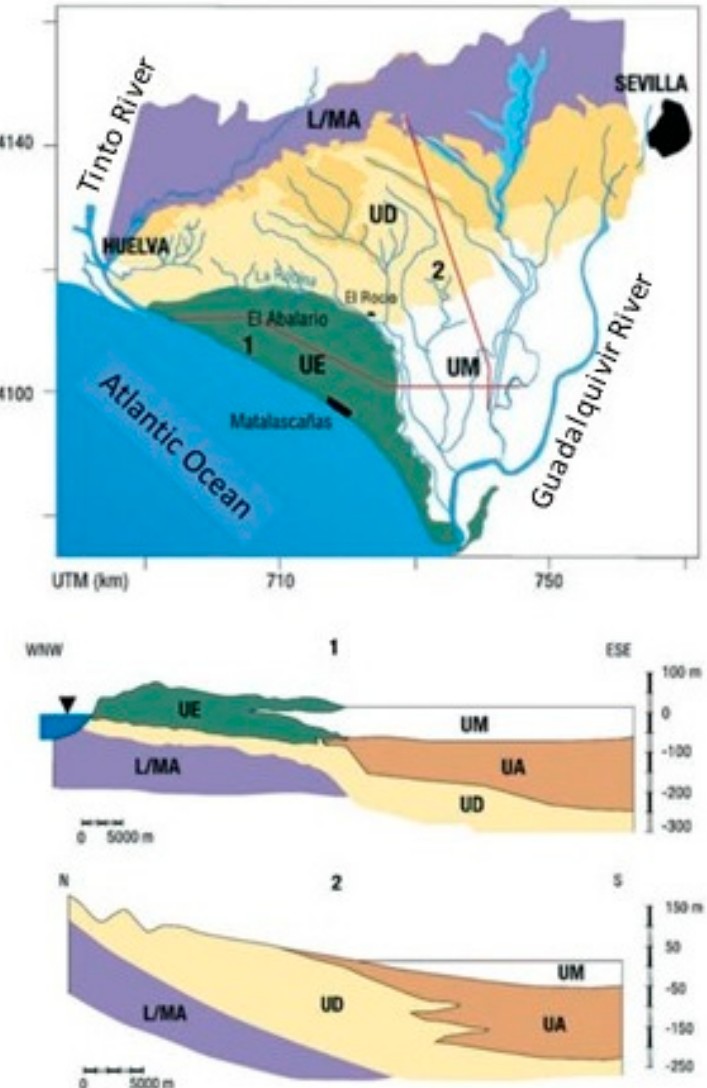

**Figure 1.** Hydrogeological sketch and structure of the Almonte–Marismas aquifer in southern Spain. The main Hydrogeological Units are as follows: L/MA stands for Miocene and Pliocene Unit (marls and silts). UD stands for Deltaic Unit (silts and sands). UM stands for Marsh Unit (clays and silts). UE stands for Aeolian Unit (fine-grained well-classified sands). UA stands for Alluvial Unit. Below, a cross section (CS) from W to E (CS 1) and from N to S (CS-2) can be seen. Modified from [29].

Regionally, the sand-dune aquifer behaves as a free aquifer which, in some places, has two layers, a coarse layer at depth and a fine-sand layer overlying it. The Marsh (clay) Unit behaves as a confined aquifer. Beneath the clays, silts and sands are saturated. Recharge to the aquifer occurs by rain infiltration and directing excess irrigation water into the agricultural areas, although they are irrigated with local groundwater coming from the same aquifer. Groundwater flows regionally to the S and SE, to the Rocina and Partido streams and to the sea. Under the natural regime, groundwater also feeds dependent ecosystems such as ponds and phreatic wetlands situated on top of the aeolian sands. The SE sector of the confined aquifer (Marsh Unit and Alluvial Unit) contains almost immobile old connate marine water, which has not been flushed out since the Holocene-sea-level stabilization [5]. Nowadays, agriculture has played a great part in the natural discharge of the aquifer. A great deal of water is pumped out from the unconfined area but near the marshes, so natural seepage to the dependent ecosystems has diminished greatly. Some local flow reversals have even been detected in the NE of the marshes, contributing to the salinization of some irrigation wells [31]. Water is abstracted from the aquifer primarily for

agriculture, particularly for growing fruits such as strawberries (now covering 6000 ha), cotton and rice and for human consumption in some coastal resorts.

　　For this paper, the study area comprises four ponds that are positioned in the sand-dune aquifer and near a coastal resort, which, during slow periods for tourism has only 3000 inhabitants, but can reach up to 150,000 in the summer months (Figure 2). This, along with nearby agricultural farms, is believed to exert a strain on the ponds in the DNP, providing a challenge to maintaining water resources while still meeting the water needs of stakeholders.

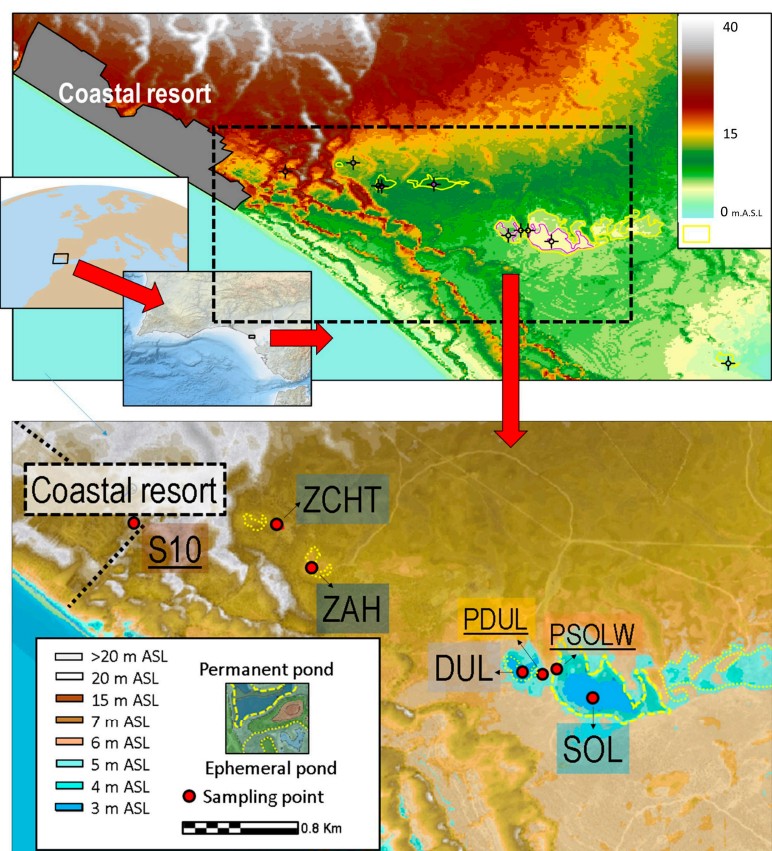

**Figure 2.** Location of Doñana in southern Spain and study area. Notice the seven main piezometers, wells and ponds studied. Legend of sites: SOL: Santa Olalla pond; PSOLW: Piezometer in Santa Olalla pond; DUL: Dulce pond; ZAH: Zahillo pond; ZCHT: Excavation made in Charco del Toro dry pond; S10: Pumping well in Matalascañas (coastal-resort area) for urban water supply.

## 3. Materials and Methods

　　To measure changes in surface-water and groundwater levels, as well as water temperature, seven high-resolution (3 h intervals) pressure transducers and water thermistors (divers and leveloggers) were installed in piezometers (two 15 m depth piezometers, PDUL and PSOL), a 150 m depth groundwater pumping well (S10), two dry ponds (ZCHT and ZAH) and two semi-permanent ponds (SOL and DUL), see Figure 2 for details. Furthermore, to calibrate temperature, a portable multimeter (HACH-HQ40D) was used during each field campaign. Measurements were recorded from May 2021 to May 2022.

　　Analysis of the hydraulic time series was performed using wavelet analysis [32–34], specifically by using the WaveletComp package in R [20], which perform these calculations. Wavelet analysis utilizes the Morlet-wavelet transform, which calculates the frequency structure across time, producing an optimal time–frequency-resolution spectrogram [19,20,23]. Morlet wavelet introduces a set of functions in the shape of small waves translated in time by, and scaled in frequency from, a simple generator function [24]. Wavelet analysis requires complete records with no gaps, so the few randomly distributed missing values

found in the time series (<1%) were imputed through linear interpolation. Furthermore, before computing the wavelet-power spectrum, the input time series are detrended, using local polynomial regression, and standardized. Both single-wavelet-transform and cross-wavelet-transform analyses have been performed. The cross-wavelet transform is the analog of the covariance, and it allows the comparing of the frequency contents of two time series and the drawing of conclusions about the synchronicity of the series at specific periods and time ranges [20]. This detailed analysis of frequency changes in water level and temperature can tell a detailed story of the interactions among groundwater, surface water, the biotic process, and anthropogenic activities. An upper period of 8 days was specified for wavelet decomposition to focus on the high-frequency components. The square root of the wavelet spectrograms are represented by a color scale to accentuate the color of the image, and the obtained figures are interpreted in the Results Section of this paper. Furthermore, visual and descriptive analysis through time-series graphs are also used to illustrate changes in hydrologic behavior.

## 4. Results

Figure 3 depicts the evolution of temperature for surface water (the DUL and SOL ponds and the ZCHT), and groundwater (the PSOL, ZAH and S10) within the DNP. Figure 3A depicts the Santa Olalla (SOL) evolution in temperature over the course of the study period. In the summer months (June–September) the temperatures of the surface water rise, mirroring the air temperature at that time of the year. The piezometer PSOL (Figure 3B) measures the groundwater temperature near Santa Olalla and registeredthe same pattern as the surface-water temperature. Dulce pond (Figure 3C) is shown to have a high frequency of changes. However, overall, it follows a similar trend to that of the surface water in Santa Olalla. The Zahillo dry pond (Figure 3D) displays similar measurements to those of the PSOL, because the piezometer PSOL also measures groundwater temperatures. The ZCHT reflects the same general pattern found in other surface-water measurements (DUL, SOL). Lastly, S10 (Figure 3F) shows a slight change in temperature in the summer months, with a drop from roughly 20.3 to 19.7 °C during summer extractions of groundwater from the aquifer.

Groundwater and surface-water levels, expressed as m above sea level, are illustrated in Figure 4A–G. Figure 4A illustrates the change in the level of Santa Olalla in relation to the pond's floor. The level begins at its highest point at the beginning of the study period and then slowly declines until December. Precipitation was low during the studied period, except for a precipitation event in October 2021 and some rainfall in January and then April. The Dulce pond, shown in Figure 4C, like Santa Olalla, shows a gradual decrease in water level, with gradual increases in October and January. The corresponding piezometers for the ponds show the same pattern, except with more intense increases in water level for the same periods (October and January). The Zahillo pond, pictured in Figure 4E, has a relatively stable water level, with only minor increases, possibly due to precipitation. It must be noted that the Zahillo Pond was dry during the studied period, as can be deduced from the position of the pond floor, which is indicated as a dotted blue line in all the ponds and piezometers. The ZCHT has an overall descending trend, with a major increase in water levels due to a major precipitation event. The well referred to as S10, see Figure 4G, mirrors the same drastic changes that are shown in the temperature diagrams, due to groundwater withdrawal to support human water consumption in the summer months. Moreover, regarding the relative position of both the water level of the ponds and the piezometric levels, (Figure 4H) the Charco del Toro pond (ZCHT) is placed at the highest altitude of 12 m asl. The Zahillo-pond water level (ZAH) is located at c. 8 m asl; the pond remained dry during all the studied period, as mentioned before. The relative position of the piezometric levels in the PDUL and PSOL (c. 6 m asl) in relation to the Dulce and Santa Olalla ponds (c. 5–5.5 m asl) shows that the water table is always above the pond level, indicating a net groundwater discharge from the aquifer to the ponds. Lastly, the S10 piezometric level is above sea level, except for the periods of pumping in

summer. The relatively fast recovery of the piezometric level to former values (c. 4.5 m asl) after the summer months indicates that the sand-dune aquifer has high transmissivity ($7.5 \times 10^{-4}$ m/s) [35].

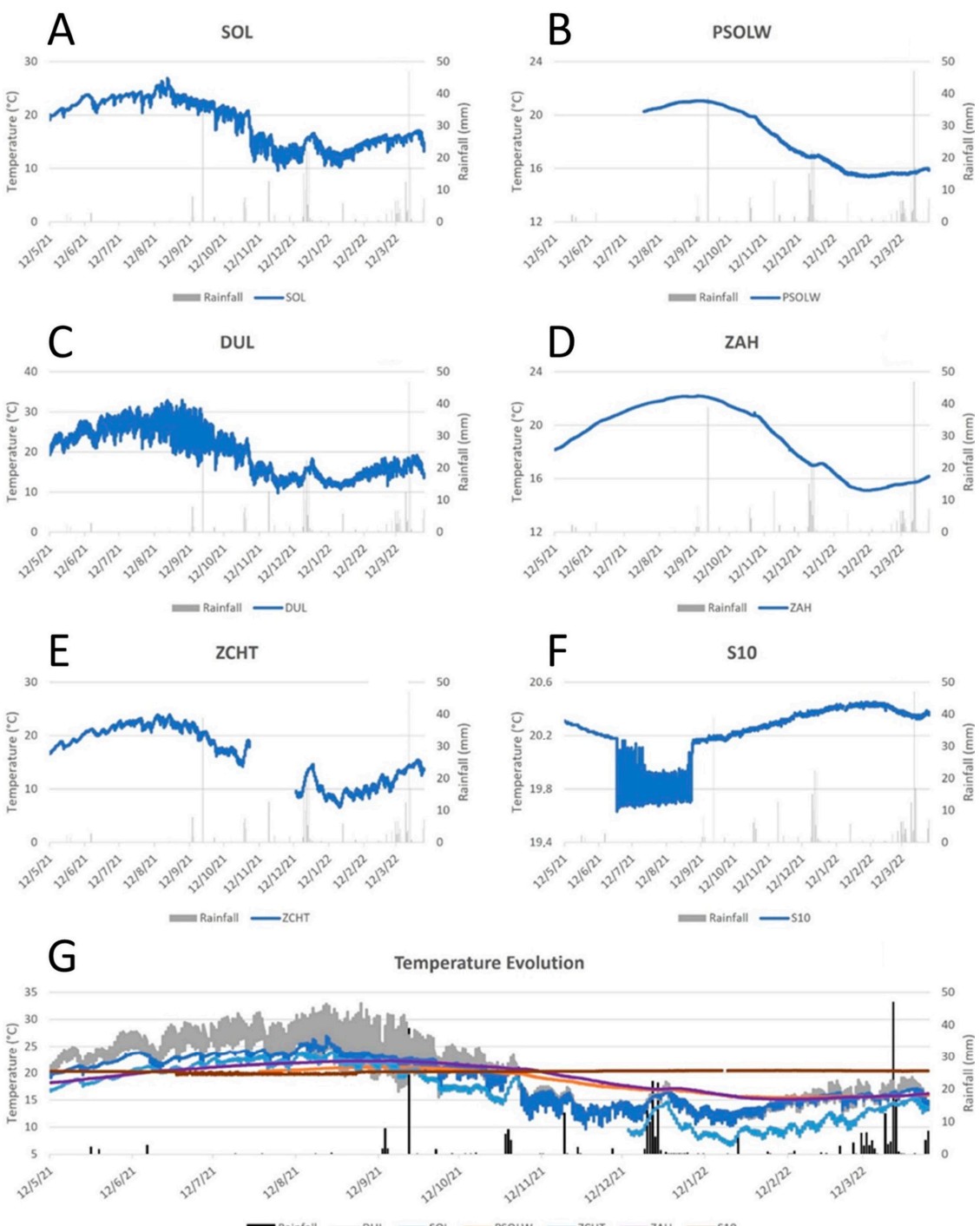

**Figure 3.** Water-temperature evolution in the ponds and piezometers. (**A**). Santa-Olalla-pond water temperature. (**B**). Santa-Olalla-groundwater temperature, registered by piezometer. (**C**). Dulce-pond water temperature. (**D**). Zahillo-pond water temperature. (**E**). Charco-del-Toro-excavation water temperature. (**F**). Sensor-S10 (located in Matalascañas for urban water supply) groundwater temperature registered by piezometer. (**G**). Temperature evolution in all the points monitored; SOL: Santa Olalla pond. PSOLW: Piezometer in Santa Olalla pond. DUL: Dulce pond. ZAH: Zahillo pond. ZCHT: Charco del Toro excavation. S10: Pumping well in Matalascañas for urban water supply.

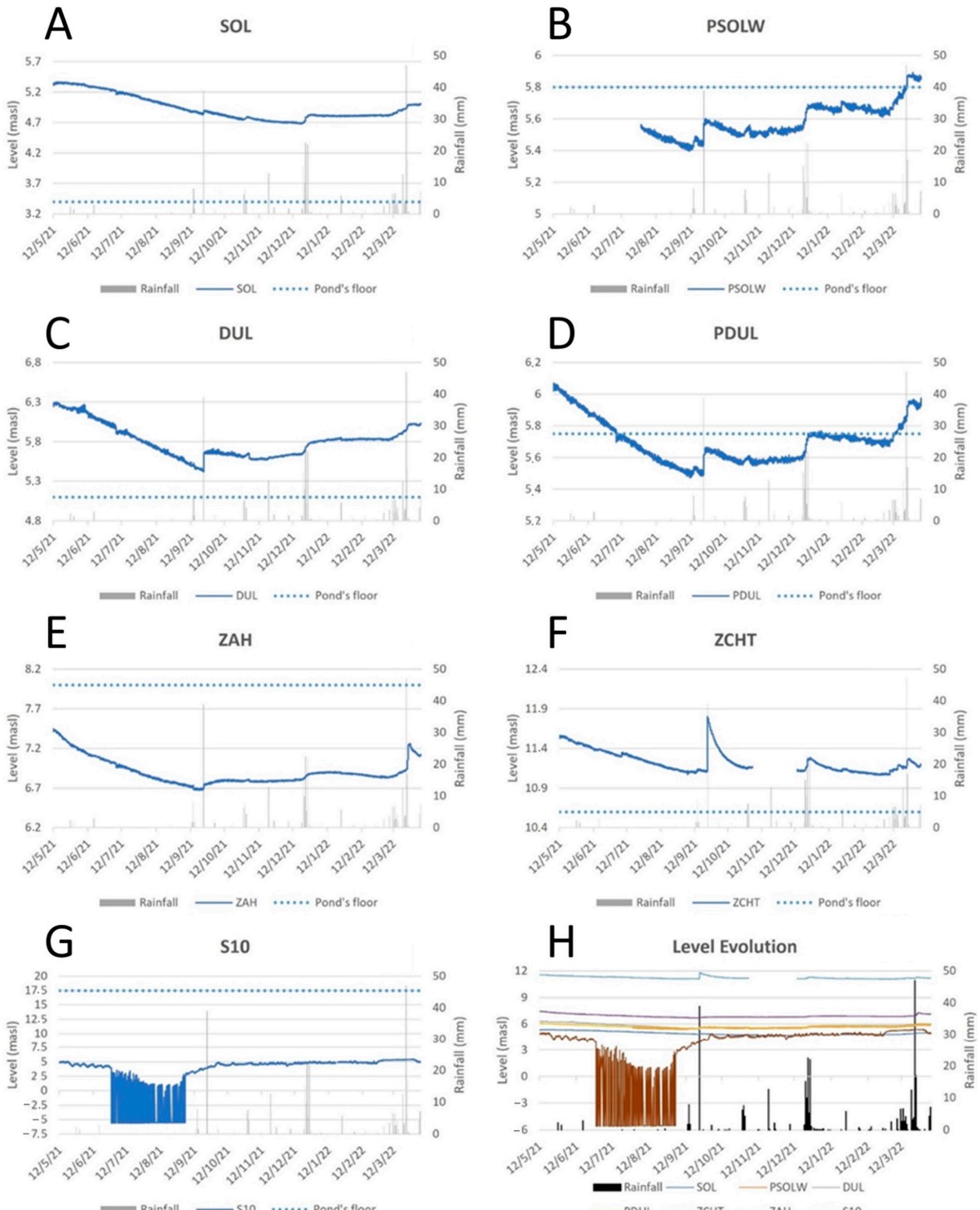

**Figure 4.** Water-level evolution in the ponds and piezometers; rainfall (mm) is represented in the secondary Y axis. (**A**). Santa-Olalla-pond water level. (**B**). Santa-Olalla groundwater level registered by piezometer. (**C**). Dulce-pond water level. (**D**). Dulce-groundwater level registered by piezometer. (**E**). Zahillo-pond water level. (**F**). Charco-del-Toro excavation water level. (**G**). Sensor-S10 (located in Matalascañas for urban water supply) groundwater level registered by piezometer. (**H**). All measurements together are represented.

Wavelet analysis of both groundwater and surface water in the DNP was carried out in all the monitored sites. Some representative examples have been chosen to exhibit the behavior of both temperature and water level; see Figure 5 (DUL and SOL) and Figure 6

(S10, PDUL and PSOLW). The color scale represents the wavelet-power spectrum of the series, with bright red areas indicating the highest levels of periodicity. The areas enclosed by white contour lines symbolize significant periodic components in the time series. For example, the Dulce and SOL ponds show daily patterns for water-temperature behavior, but during different time spans. This could be related to the flooded area (i.e., the volume of water stored), but also to differential SW–GW interactions in both ponds. Moreover, in the Dulce level (Figure 5A), there is a consistent pattern of red at 0.5 days, which symbolizes the fact that there is a strong 0.5-day periodicity in the series for the corresponding time span.

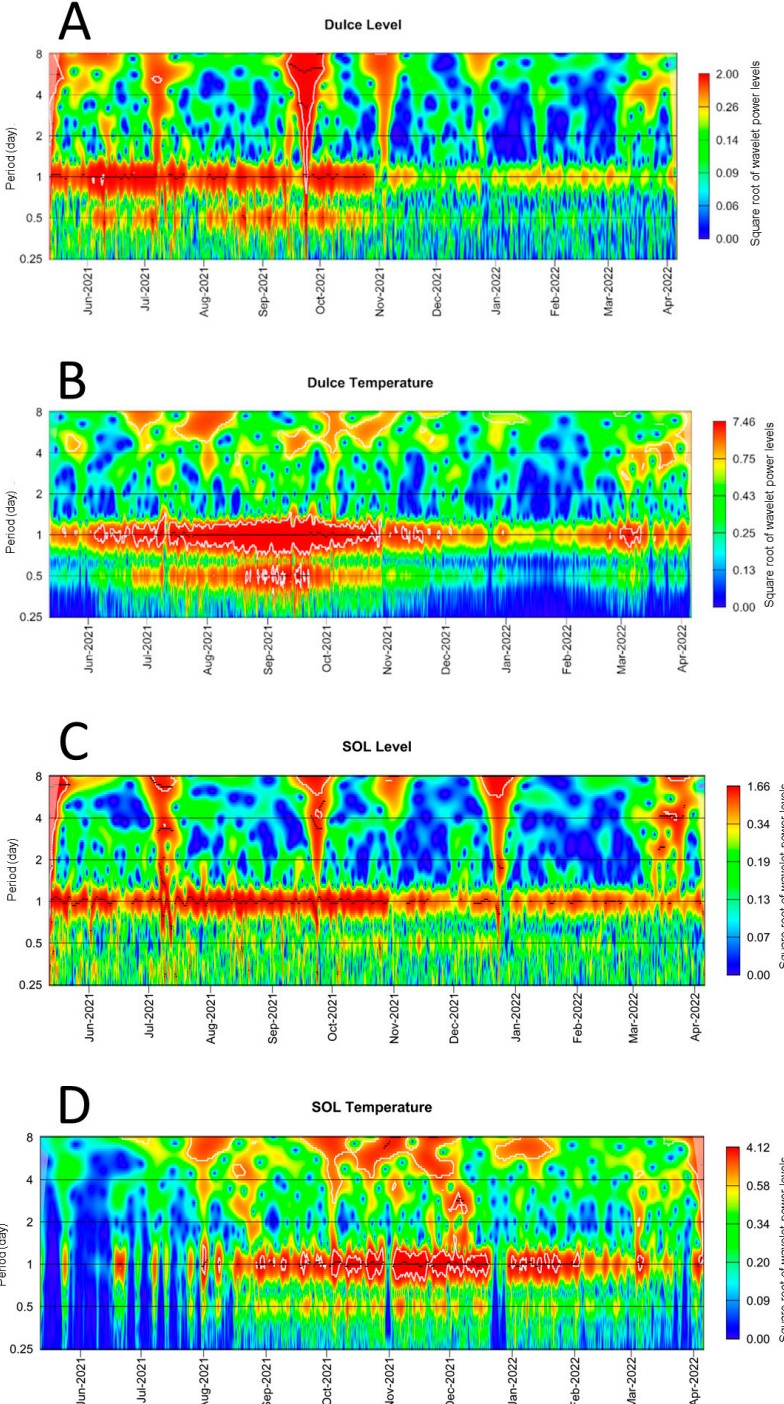

**Figure 5.** Wavelet spectrograms. (**A**). Water-level wavelet analysis in Dulce pond. (**B**). Temperature wavelet analysis in Dulce pond. (**C**). Water-level wavelet analysis in SOL pond. (**D**). Temperature wavelet analysis in SOL pond.

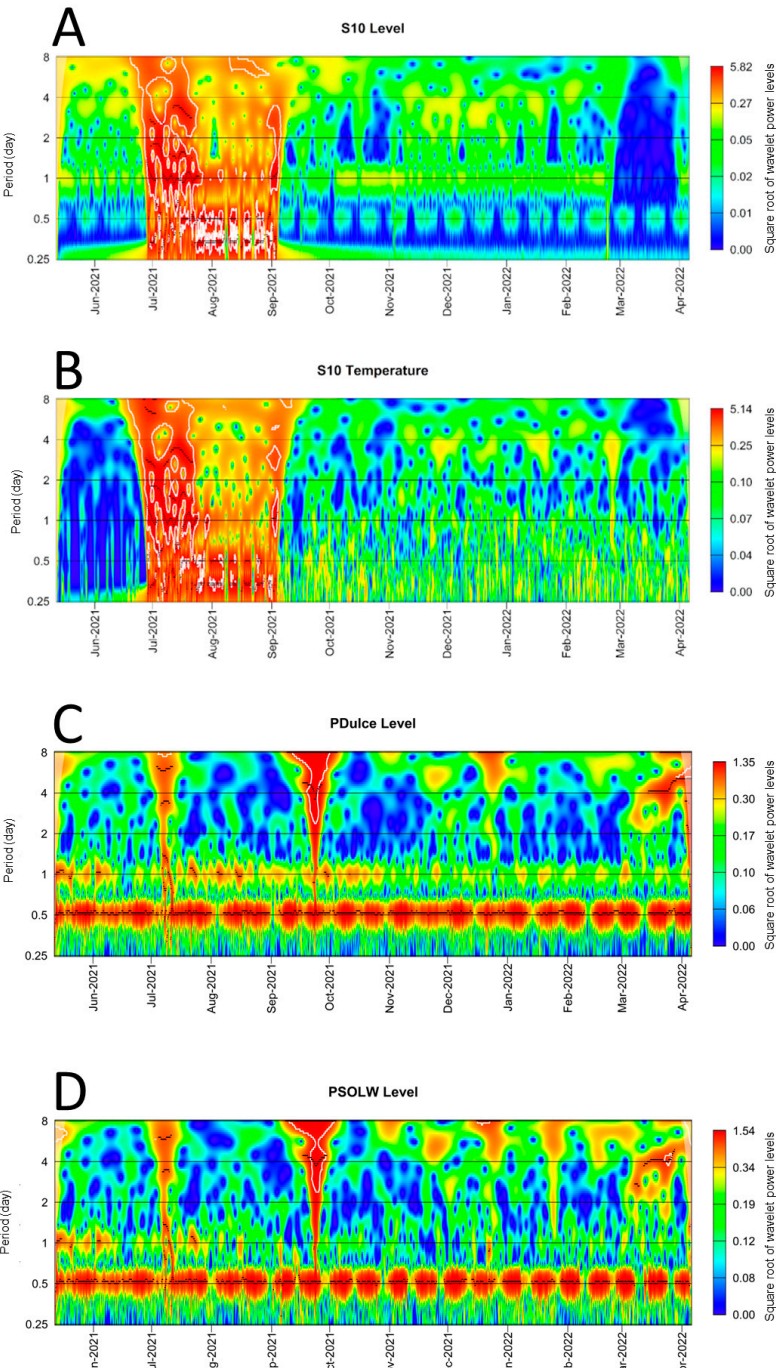

**Figure 6.** Wavelet spectrograms. (**A**). Water-level wavelet analysis in S10. (**B**). Temperature wavelet analysis in S10. (**C**). Groundwater-level wavelet analysis in PDUL piezometer. (**D**). Groundwater-level wavelet analysis in PSOLW piezometer.

Therefore, this shallow ecosystem shows profound water-level oscillations every 0.5 days. We have attributed this pattern as reflective of the tidal processes affecting the levels of the groundwater (Figure 6C). Other possible causes such as sub-daily evaporation fluxes correlated to wind speed and vapor-pressure deficit found by Fournier et al. (2021) [36] in boreal reservoirs are unlikely in this context, due to shallowness. Fernández-Ayuso et al. (2019) [15] previously detected the effect of semi-diurnal tides in groundwater in this area in the medium-to-deep-groundwater levels. However, its detection in shallow groundwater (ZAH) and even surface water (DUL) is a new finding. The PDUL exhibits the same effect, as can be seen in Figure 6C.

For the S10 (Figure 6A), in the months of July to September, there are extremely noisy variations displayed, due to summer-groundwater abstractions in the coastal resort. Drops in the piezometric level of more than 20 m registered by the sensor installed inside the drill, almost instantaneously, and a subsequent and fast recovery of the level when the pumping stops, is the reason why wavelet results from July to September are numerical artifacts. As explained, due to the activation/deactivation of the pump in the summer months, when the tourists arrive at the coastal resort, the oscillation in the water level is irregular. For the temperature, the explanation is similar, because the drop in the water level leaves the sensor above the saturated zone inside the drill, so an abrupt change in temperature takes place (see Figure 3F).

In Figure 5C (SOL level) it can also be seen that there is a strong daily pattern, with changes in the water level. We have attributed this to the daily rising and lowering of the water level of the pond, due to natural processes such as evaporation and evapotranspiration. When we look at the Dulce pond level in Figure 5A, there is a significant change in the daily level and a faint effect at 0.5 days (the tidal effect). Nonetheless, these are preliminary data, so, in the future, longer time series will be analyzed. To elaborate further on this point, when investigating this area there was the potential that the groundwater and surface water could be independent systems; for example, there could be a pond unrelated to the regional groundwater system but sustained by groundwater discharge from a perched aquifer (Figure 7C,D). Alternatively, we could have a system directly connected to the regional aquifer (Figure 7A,B). The fact that surface-water measurements also display effects from the tides proves that the two systems are connected to the regional aquifer. The key takeaway from this conclusion is that we can see through the wavelet analysis that the shallow groundwater level in DNP, but also the surface-water level in some ponds (e.g., the Dulce pond) are influenced by tidal processes.

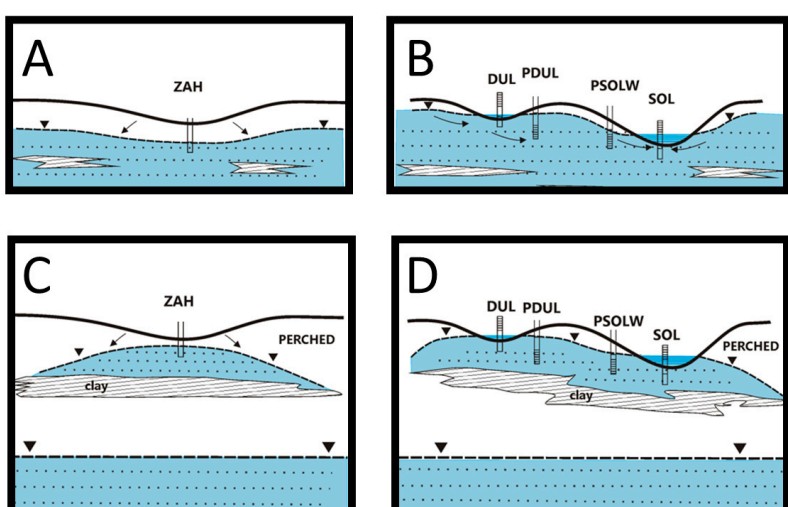

**Figure 7.** Hypothesis of the hydrogeological functioning of ZAH, SOL and DUL ponds based on the results obtained. (**A**). ZAH pond is dry but connected to the coastal aquifer. (**B**). DUL and SOL ponds are flooded and connected to the coastal aquifer. (**C**). ZAH pond is dry but disconnected from the coastal aquifer. (**D**). DUL and SOL ponds are flooded and disconnected from the coastal aquifer. Cases (**A,B**) (connection of ZAH, DUL and SOL ponds to the coastal aquifer) are the ones supported by the results obtained in this investigation.

To further analyze the variables water-level and water temperature, a cross-wavelet analysis was applied in the studied ponds. Several authors have pointed out the interest and importance of the application of this type of analysis in hydrological studies [37]. The cross-wavelet analysis can reveal the correlation degree between level and temperature variations on the daily and sub-daily scales [24]. It can be seen how temperature and water level are coherent in time and frequency. In Figure 8, the cross-spectrograms of

the Dulce, Santa Olalla and Zahillo pond systems can be observed. Surface-water level and temperature show strong daily joint periodicity, especially during summer and early autumn, relating to the evaporative cycles. Horizontal arrows pointing to the right indicate that the two series are in phase, with vanishing phase differences (i.e., synchronized), especially for the Dulce pond. These arrows are plotted only within white contour lines, indicating significance (with respect to the null hypothesis of white-noise processes) at the 5% level. However, on shallow groundwater, level and temperature do not exhibit significant spectral-wavelet correlation, although the highest cross-wavelet power in Santa Olalla belongs to a 0.5-day period throughout the entire series.

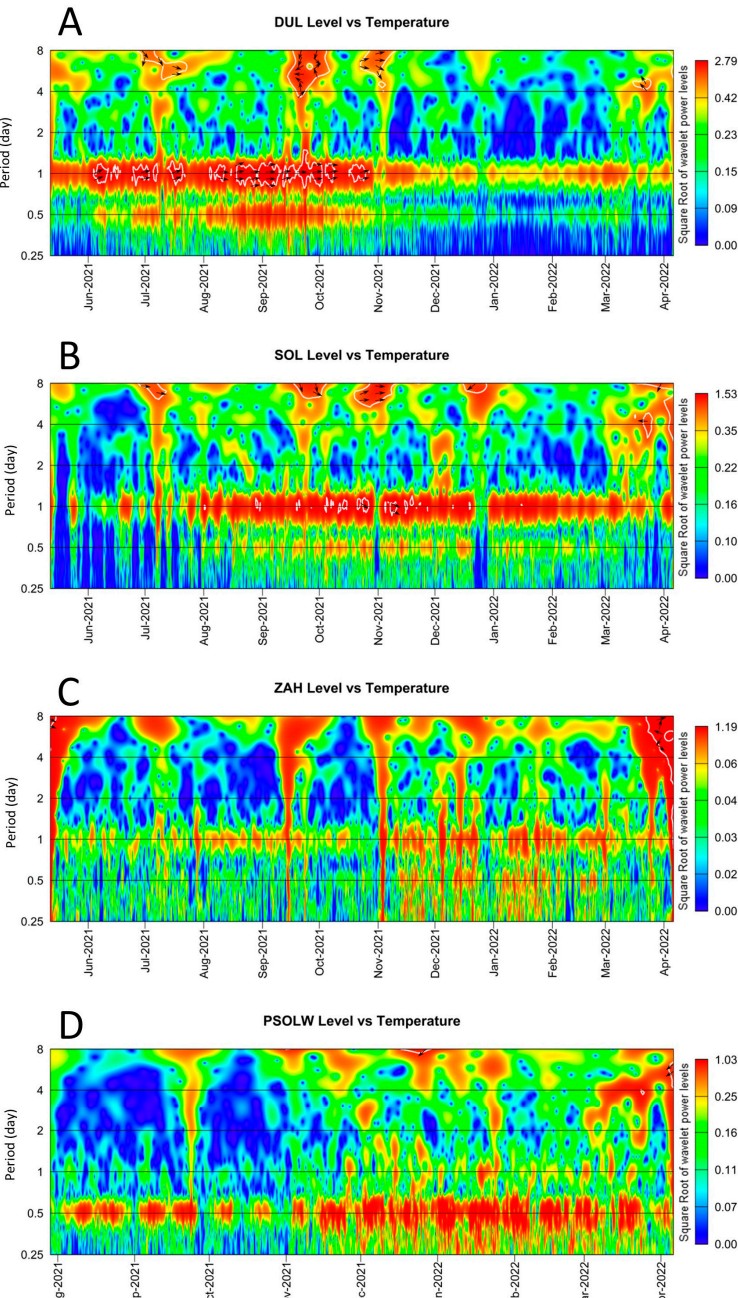

**Figure 8.** Cross-wavelet spectrograms. (**A**). Dulce Pond (DUL) surface-water-level water-temperature cross-wavelet spectrogram. (**B**). Santa Olalla (SOL) surface-water-level water-temperature cross-wavelet spectrogram. (**C**). Zahillo Pond (ZAH) surface-groundwater-level groundwater-temperature cross-wavelet spectrogram. (**D**). Santa-Olalla-Pond (PSOLW)-piezometer groundwater-level groundwater-temperature cross-wavelet spectrogram.

## 5. Discussion

Many authors have established a clear effect of groundwater pumping in the coastal resort to several sand-dune ponds of Doñana, some of them included in this study, e.g., the Zahíllo pond—ZAH—and the Charco del Toro pond—ZCHT [38]. On the other hand, despite the requests by UNESCO to cease banned groundwater withdrawal for irrigation in the north of the DNP, the problem is yet far from being solved. Measures taken by the administration, such as the closing of illegal drills, or the public acquisition of a large property (called *Mimbrales* farm) in 2015 to save more than 6.8 hm/y in irrigation-water rights [2], are initiatives that point in the right direction, but they are not sufficient to solve the problem. The effect of aquifer exploitation is obvious in the disappearance of such dune ponds, which depend on the water table and hold endemic species and rich communities of amphibians and insects. In addition, on 9 February 2022, the Andalusian regional parliament voted to support a plan to legalize 1500 ha of irrigated land and thus legitimize the operations of illegal farmers, despite open opposition from the Spanish central government, EU, UNESCO and several nongovernmental organizations [39]. Some studies on the detailed hydrology of the ponds located in the Abalario sector of DNP [40–42] explain the inundation phase of the systems as being disconnected from the regional dune aquifer of Doñana. In this sense, they establish the genesis and formation of the ponds to be related to the autogenic or autogenous formation of a shallow organic-clay layer of very low permeability during high-water-level periods. The formation of such a layer leads to the origin of a perched aquifer, unrelated to the regional aquifer, as can be viewed in the sketch of Figure 7C,D (e.g., the ZAH, DUL and SOL ponds). Although clay lenses are not inexistent in the area, as they are a common feature in this sedimentological context, the hypothesis of these authors [40–42] is not supported by the results obtained in this study. In the hypothesis of [40–42], consistent with a perched aquifer, the surface-water level and piezometric oscillation in the groundwater near the ponds would not be influenced by the tidal oscillation affecting the regional coastal aquifer. Another possibility, which may be coherent for the ponds studied in this paper, would be that the dry ponds (e.g., the ZAH) were perched systems—for that reason they dried out—and the permanent ones (e.g., the SOL) are ponds connected to the coastal aquifer. Again, the result of this study reveals that both dried-out ponds (the ZAH) and permanent ponds (the SOL) are indeed connected to the aquifer. This is coherent with the results obtained in previous studies, proving a direct connection between the regional groundwater system and some of the studied ponds (e.g., [15]).

Olías-Alvarez [43] detected for the first time semi-diurnal tides in the DNP, in a piezometer at 6 km from the sea, with an amplitude of only 1 cm, much smaller than the amplitude measured by Fernández-Ayuso and Rodríguez-Rodríguez [44] in the PSOL and PDUL, of 2.5 cm, which seems to indicate that the tidal signal attenuates with respect to the distance to the ocean in our study case.

The literature shows other examples of the detection of semi-diurnal tides in surface water. On the Yucatan Peninsula, a karstified-limestone area, some authors [45] found the tidal effect even in the water bodies more distant from the sea, and proved that there was a clear connection between such water bodies and the regional aquifer that were previously assumed to be hydrologically isolated from the regional aquifer. The tidal sign was also detected in shallow groundwater in ephemeral groundwater-fed lakes (turloughs) in Ireland [46], even at an 8 km distance from the sea, in a karstic-hydraulic system, much less transmissive than the aeolian sands in our study site.

Finally, the fact that shallow-groundwater temperature and level exhibit a correlation at a 0.5 day period in the Santa Olalla pond can be attributed to dampened-temperature variations caused by the force of tidal fluctuations driving heat transport by advection and dispersion, as stated by other authors [47–49]. However, this will require further analysis, as it was not detected at the Dulce pond.

## 6. Conclusions

Through wavelet analysis and visual interpretation, we were able to provide evidence of a connection between surface-water and regional-groundwater systems in sand-dune ponds of DNP, notwithstanding the hydrological affect due to groundwater withdrawal. The data series contained temperature and water-level information in a 3 h time step of two shallow piezometers, one deep pumping well and four ponds, from May 2021 to May 2022. Such ponds are groundwater-dependent ecosystems of a through-flow type, and therefore are not perched over the saturated zone, as incorrectly stated by previous authors. From the evidence of semi-diurnal tides affecting not only the groundwater measurements (sand-dune regional aquifer), but also the surface-water measurements (shallow ponds), and temperature, we were also able to detect other processes, such as evaporation or evapotranspiration. The results of this study have contributed to the improvement in the current knowledge of the hydrological behavior of shallow water bodies. It is worth highlighting the fact that it is the first time that the effect of the Doñana sand-dune aquifer on the water-level oscillation of a shallow pond (c. 0.5 m depth) has been detected at an hourly interval. There were no other patterns to note in the water-level and temperature data sets of the studied ecosystems. None the less, we encourage water authorities to continue to monitor the situation so we may maintain this precious ecosystem.

**Author Contributions:** M.R.-R. and J.T. conceived of the presented idea. H.A., A.F.-A. and J.T. developed the theory and performed the computations. M.R.-R. and M.J.M.-V. verified the analytical methods. N.F.-N. aided with the elaboration of the final version of the manuscript and the elaboration of several figures. All authors discussed the results and contributed to the final manuscript. All authors have read and agreed to the published version of the manuscript.

**Funding:** This research was funded by the collaboration agreement between the Guadalquivir Hydrographic Confederation and the Pablo de Olavide University (UPO): "Study of hydrological monitoring and modeling of the pond-aquifer relationship in the Doñana aquifer. Inventory and monitoring" and the FEDER project UPO-1259543.

**Data Availability Statement:** Data sets collected for this work are not published in any repository. Due to the nature of the research, due to confidential uses supporting data is not available.

**Acknowledgments:** This work has been possible thanks to the collaboration agreement be-tween the Guadalquivir Hydrographic Confederation and the Pablo de Olavide University (UPO): "Study of hydrological monitoring and modeling of the pond-aquifer relationship in the Doñana aquifer. Inventory and monitoring" and the FEDER project UPO-1259543. The logistical and technical support of the Singular Scientific-Technical In-frastructure of the Doñana Biological Reserve (DBR-ICTS) is gratefully acknowledged.

**Conflicts of Interest:** The authors declare no conflict of interest.

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
