# Peer review of "Wavelet Analysis on Groundwater, Surface-Water Levels and Water Temperature in Doñana National Park (Coastal Aquifer in Southwestern Spain)"

_water, doi:10.3390/w15040796_

Round 1
Reviewer 1 Report
Rephrase/Revise the results section for figure 4 (A- D) and figure 5(A-D) for more clarity and to understand the the wavelet analysis.
Add some more reviews to support the results in discussion section.
Author Response
REVIEWER 1:
We appreciate the reviewer comments and suggestions, definitely leading to an improved version of our manuscript.
Rev1-C1: Rephrase/Revise the results sections for figure 4 (A-D) and figure 5 (A-D) for more clarity and to understand the wavelet analysis.
Rev1-C1Response: Thanks. The y-axis was inserted in new Fig. 4 and Fig.5, and the color bar label of the wavelet power spectrum inverted in the new version of the MS. Also, the section has been revised and we have added more information in lines 369-399.
Rev1-C2: Add some more reviews to support the results in the discussion section.
Rev1-C2Response: Thank you. Discussion section was improved and more information between lines 356-372 is included:
“Olías-Alvarez [46] detected for the first time semi-diurnal tides in the DNP, in a piezometer at 6 km from the sea, with an amplitude of only 1 cm, much smaller than the amplitude measured by Fernández-Ayuso and Rodríguez-Rodríguez [47] in PSOL and PDUL, of 2 m, which seems to indicate that the tidal signal attenuates with respect to the distance to the Ocean in our study case.
Literature shows other examples of the detection of semi-diurnal tides in surface water. In Yucatan Peninsula, a karstified limestone area, some authors [48] found the tidal effect even in the water bodies more distant to the sea and proved that there was a clear connection between such water bodies and the regional aquifer, previously assumed to be hydrologically isolated from the regional aquifer. The tidal sign was also detected in shallow groundwater in ephemeral groundwater-fed lakes (turloughs)in Ireland [49], even at 8 km distance to the sea, in a karstic hydraulic system, much less transmissive than the aeolian sands in our study site.
Finally, the fact that in shallow groundwater temperature and groundwater level variables exhibit correlation at 0.5 days period can be attributed to dampened temperature variations caused by the force of tidal fluctuations driving heat transport by advection and dispersion, stated by other authors [38-40]. “
Reviewer 2 Report
Reviewer’s Report on the manuscript entitled:
Wavelet analysis on groundwater, surface water levels and water temperature in Doñana National Park (Southwestern Spain)
The authors investigate the potential of wavelet transform for analyzing water level and temperature for Donana Nationa Park. Though topic and methods are interesting, the manuscript should be re-worked, and longer time series (decades) should be considered. Furthermore, cross-wavelet transform, and wavelet coherence should be considered for studying the impact of temperature on water level when using wavelets. Please see my comments below.
General comments.
Line 55-62. You need to at least at one paragraph to describe wavelet analysis and its applications in hydrology. For example, the least-squares wavelet software applied to water discharge, temperature, and precipitation:
https://doi.org/10.1016/j.ejrh.2021.100847
https://doi.org/10.1007/s10291-019-0841-3
Or continuous wavelet transform and wavelet coherence applied to rainfalls and run-off:
https://doi.org/10.3390/w13223243
https://doi.org/10.1016/j.aej.2020.01.050
Furthermore, Please add the cross-spectrogram using cross-wavelet transform or the least-squares cross-wavelet transform for water level and temperature along with their phase delays (displayed by arrows on the cross-spectrograms). This way, one can clearly see how the temperature and water level are coherent in time and frequency. See the articles above for more information.
Line 121. Please also include the first reference above here for further explaining the use of Morlet wavelet which produces an optimal time-frequency resolution spectrogram.
Please insert the y-axis unit in Figures 4 and 5. Is it Period (years)? Also, the color bar label should be oriented 180 degrees to be written from bottom to top not top to bottom. You should display the related time series above each spectrogram. See the first reference above as an example. Also, one year of data cannot show the inter-annual cycles (between years) so the periods higher than 1 are not reliable using one year of data (see the cone of influence for the continuous wavelet transform). Longer time series are extremely useful for such analysis definitely not one year!
Regards,
Author Response
REVIEWER 2:
Rev2-C1. The authors investigate the potential of wavelet transform for analysing water level and temperature for Doñana National Park. Though topic and methods are interesting, the manuscript should be re-worked, and longer time series (decades) should be considered. Furthermore, cross-wavelet transform, and wavelet coherence should be considered for studying the impact of temperature on water level when using wavelets. Please see my comments below.
Rev2-C1Response. We appreciate the reviewer’s comments and suggestions, definitely leading to an improved version of our manuscript. It seems we hadn’t explained ourselves clearly concerning the use of longer time series. Overlapping level and temperature time series for all the coastal ponds are only available for one year. However, these are high-resolution series (3-hour intervals), which allows us to study high-frequency (e.g., daily and sub-daily) components in the time-frequency domain through continuous wavelet analysis. From this analysis we can’t infer general characteristics of the hydrological functioning, for which we would need decadal data to detect inter-annual components as the reviewer suggests, but it can provide insights and contribute to the debate of the degree of connectivity between surface and groundwater systems in Doñana National Park.
As described below, we have now made these points clear in the paper, and also added the cross-wavelet transform between temperature and water level.
General comments.
Rev2-C2. Line 55-62. You need to at least at one paragraph to describe wavelet analysis and its applications in hydrology. For example, the least-squares wavelet software applied to water discharge, temperature, and precipitation:
https://doi.org/10.1016/j.ejrh.2021.100847
https://doi.org/10.1007/s10291-019-0841-3
Or continuous wavelet transform and wavelet coherence applied to rainfalls and run-off:
https://doi.org/10.3390/w13223243
https://doi.org/10.1016/j.aej.2020.01.050
Rev2-C2Response. Thanks. We ha added the following lines on wavelet analysis and its applications, as well as the suggested references (lines 74-96):
“Wavelet Analysis used in the field of hydrology has been applied since the 1990s for multi-temporal scale analysis of hydrometeorological series, with increasing attention in the last decade [Sang, 2013; Schulte et al., 2016; Ghaderpour et al., 2021]. Although most studies focus on the analysis of long climatic and hydrological time series to reveal inter-annual components [Ghaderpour et al., 2021; Zmarane et al., 2021], it has been shown (Fernández-Ayuso et al., 2019) that this methodology is also helpful for shorter time series with a higher sampling frequency (e.g., hourly). The interpretation of the hydrological processes in a given water body (e.g., river, pond, playa-lake, aquifer, etc.) derived from a high temporal resolution but short (e.g., annual) time series will be different from a low temporal resolution but long time series (e.g., decadal). For this particular study, we have used a high-resolution (3-hour) yearlong time series to interpret the hydrological behavior of a series of ponds. The main reasons for this approach are: Firstly, the authors made a previous study in the same area but focused on the wavelet analysis of piezometric levels during a 2-year timespan [Fernández-Ayuso et al., 2019] to enhance the hydrogeological conceptual model in one particular permanent pond, not affected by anthropic disturbances (Fig. 1, SOL pond). However, here, the focus has been extended to four ponds, some of them affected by anthropic disturbances [Fernández-Ayuso et al., 2018], where surface and groundwater level and temperature time series have been compared. Secondly, for the first time, one of the ponds studied (DUL, see Fig.1) has been monitored with this high temporal resolution. Finally, the general hydrogeological behavior of the unconfined sand dune aquifer has been previously described, but not the interactions of this aquifer with the sand dune ponds included in this paper.”
Rev2-C3: Furthermore, Please add the cross-spectrogram using cross-wavelet transform or the least-squares cross-wavelet transform for water level and temperature along with their phase delays (displayed by arrows on the cross-spectrograms). This way, one can clearly see how the temperature and water level are coherent in time and frequency. See the articles above for more information.
Rev2-C3Response. Thanks. A new Figure 7 has been added in the new version of ths MS. This figure shows Cross-spectrograms in Dulce - Santa Olalla hydrological system. Upper diagrams (light blue) show Dulce and Santa Olalla (SOL) surface water cross correlograms. Lower diagrams (red) show groundwater cross correlograms. “PDUL” stands for Piezometer in Dulce pond and “PSOLW” stands for Piezometer in Santa Olalla pond (Western shore).
This figure has been commented in the text (304-314):
“To further analysis of the variables water level – water temperature, cross wavelet analysis has been applied in the studied ponds. Several authors have pointed out the in-terest and importance of the application of this type of analysis in hydrological studies [GG]). The cross wavelet can reveal the correlation degree between level and tempera-ture variations at daily and sub-daily scales. This way, it can be seen how temperature and water level are coherent in time and frequency. In figure 7, the cross-spectrograms of the Dulce -Santa Olalla hydrological system can be observed. Surface water level and temperature show strong daily joint periodicity, especially during summer and early autumn, related to evaporative cycles. However, on shallow groundwater, temperature and level exhibit consistent spectral wavelet correlation at a 0.5 days period throughout the entire series.”
Rev2-C4: Line 121. Please also include the first reference above here for further explaining the use of Morlet wavelet which produces an optimal time-frequency resolution spectrogram.
Rev2-C4Response: Thanks. We have rephrased the sentence to further explain the use of the Morlet wavelet transform and added the suggested reference (172-175):
“Wavelet analysis utilizes the Morlet wavelet transform, which calculates the frequency structure across time, producing an optimal time-frequency resolution spectrogram [19-20, 23]. Morlet wavelet introduces a set of functions in the shape of small waves translated in time by and scaled in frequency from a simple generator function [24].”
Rev2-C5: Please insert the y-axis unit in Figures 4 and 5. Is it Period (years)? Also, the color bar label should be oriented 180 degrees to be written from bottom to top not top to bottom. You should display the related time series above each spectrogram. See the first reference above as an example. Also, one year of data cannot show the inter-annual cycles (between years) so the periods higher than 1 are not reliable using one year of data (see the cone of influence for the continuous wavelet transform). Longer time series are extremely useful for such analysis definitely not one year!
Rev2-C5Response: Thanks. The y-axis was inserted in new Fig. 4 and Fig.5, and the color bar label of the wavelet power spectrum inverted in the new version of the MS.
Regarding the use of 1-year data, please, see our answer to the first comment. Nevertheless, the periods shown in the Y-axis are in days, not years.
Reviewer 3 Report
Specific comments:
1. Line 16-18: Authors should point out clear if the water resources and the unsustainable extraction of water here are groundwater?
2. The whole abstract should be reorganized as the present version is too poor. Novelty of the present research should be pointed out in the manuscript. It is hard to find any novelty from the present abstract. Quantitative findings are recommended to be putted in the abstract.
3. The title did not present the distinguishing feature of the present manuscript, for example the coastal feature.
4. Do not only concern the present study area and the local interests in the introduction. Authors should consider what could the present manuscript bring to the international community. Thus, it is better to consider the present research as a kind of issue rather than a case. The introduction should review the research progress in this field.
5. Figure 1 should be involved in the section 2 rather than section 3.
6. Hydrogeological map, at least a cross section one, should be involved in the section 2.
7. It is hard to understand the hydrogeological setting of the study area without hydrogeological figure.
8. The figures should be redrawn. Authors are recommended to refer to the instruction provided by the journal.
9. The method introduce is too brief.
10. The discussion should be made much deeper.
Author Response
REVIEWER 3
Specific comments:
Rev3-C1:Line 16-18: Authors should point out clear if the water resources and the unsustainable extraction of water here are groundwater?
Rev3-C1Response: The Doñana National Park (DNP) is a protected area with water resources drastically diminishing due to the unsustainable extraction of groundwater for agricultural irrigation and human consumption of a nearby coastal city.
Rev3-C2:The whole abstract should be reorganized as the present version is too poor. Novelty of the present research should be pointed out in the manuscript. It is hard to find any novelty from the present abstract. Quantitative findings are recommended to be putted in the abstract.
Rev3-C2Response: We acknowledge the reviewer suggestion. Indeed, the original Abstract lacks relevant information about the main findings of this paper. In the new version of the MS, a new paragraph has been added. In this paragraph, the new findings of this research have been included. For example, the 6-h interval signal, related to the Atlantic Ocean tides affecting the coastal aquifer and the 12-h interval signal, related to evaporation and evapotranspiration hydrological processes have also been included. The importance of this study in the understanding of the behavior of these ponds has also been mentioned in the new Abstract (Lines 16-34):
The Doñana National Park (DNP) is a protected area with water resources drastically diminish-ing due to unsustainable extraction of water for agricultural irrigation and human consumption of a nearby coastal city. In this study, we explore the potential of wavelet analysis applied to high temporal resolution groundwater and surface water time series of temporary coastal ponds in the DNP. Wavelet analysis was used to measure the frequency of changes in water levels and water temperature, both crucial to our understanding of complex hydrodynamic patterns. Results show that the temporary ponds are groundwater-dependent ecosystems of a through-flow type and are still connected to the sand dune aquifer, regardless their hydrological affection due to groundwater withdrawal. These ponds, even those most affected by pumping in nearby drills, are not perched over the saturated zone. This was proven by the evidence of a semi-diurnal (i.e., 6-hour) signal in the surface level time series of the shallow temporary ponds. This signal is, at the same time, related to the influence of the tides affecting the coastal sand dune aquifer. Finally, we detected other hydrological processes that affectthe ponds, such as evaporation and evapotranspiration, with a clear diurnal (12-hour) signal. The maintenance of the ecological values and services to the society of this emblematic wetland is currently at jeop-ardy due to the affection of the groundwater abstraction for irrigation. The results of this study contribute to the understanding of the behavior of these fragile ecosystems of DNP, and will also contribute to a sound integrated water resources management.” *Changes are underlined.
Rev3-C3: The title did not present the distinguishing feature of the present manuscript, for example the coastal feature.
Rev3-C3Response: Thank you. The title was changed to: “Wavelet analysis on groundwater, surface water levels and water temperature in Doñana National Park (coastal aquifer in Southwestern Spain) “
Rev3-C4: Do not only concern the present study area and the local interests in the introduction. Authors should consider what could the present manuscript bring to the international community. Thus, it is better to consider the present research as a kind of issue rather than a case. The introduction should review the research progress in this field.
Rev3-C4Response: Thanks. In the introduction section, a new paragraph has been added. The novelty and justification of this work have been explained in detail. In 2019, the authors published a paper in Groundwater (Fernández-Ayuso et al., 2019). This paper focused in the wavelet analysis of piezometric levels during a two-year period in Santa Olalla pond (Fig. 1, SOL pond) to enhance the hydrogeological conceptual model. This pond was not affected by anthropic disturbances. In this new paper, the focus has been extended to four ponds, some of which show anthropic disturbances. Surface water time series (levels and temperatures) have been compared to piezometric water level/temperature time series (Lines 74-96):
“Wavelet Analysis used in the field of hydrology has been applied since the 1990s for multi-temporal scale analysis of hydrometeorological series, with increasing attention in the last decade [21-23]. Although most studies focus on the analysis of long climatic and hydrological time series to reveal inter-annual components [23-24], it has been shown [15] that this methodology is also helpful for shorter time series with a higher sampling frequency (e.g., hourly). The interpretation of the hydrological processes in a given water body (e.g., river, pond, playa-lake, aquifer, etc.) derived from a high temporal resolution but short (e.g., annual) time series will be different from a low temporal resolution but long time series (e.g., decadal). For this particular study, we have used a high-resolution (3-hour) yearlong time series to interpret the hydrological behavior of a series of ponds. The main reasons for this approach are: Firstly, the authors made a previous study in the same area but focused on the wavelet analysis of piezometric levels during a 2-year timespan [15] to enhance the hydrogeological conceptual model in one particular permanent pond, not affected by anthropic disturbances (Fig. 1, SOL pond). However, here, the focus has been extended to four ponds, some of them affected by anthropic disturbances [14], where surface and groundwater level and temperature time series have been compared. Secondly, for the first time, one of the ponds studied (DUL, see Fig.1) has been monitored with this high temporal resolution. Finally, the general hydrogeological behavior of the unconfined sand dune aquifer has been previously described, but not the interactions of this aquifer with the sand dune ponds included in this paper..”
Rev3-C5: Figure 1 should be involved in the section 2 rather than section 3.
Rev3-C6: Hydrogeological map, at least a cross section one, should be involved in the section 2.
Rev3-C7: It is hard to understand the hydrogeological setting of the study area without hydrogeological figure.
Rev3-C5,C6 and C7 Response: Thank you for the suggestion. In thiscase, the authors consider that the study area is very good described in other works cited (examples below) and in order to not be repetitive we decided to focus in the area of the specifical study that we did. The major features of the hydrogeological setting relevant for our study are described in the study site section, which has been extended.
References:
- Montes-Vega, M.J.; Rodríguez-Rodríguez, M. Análisis del hidroperíodo de tres lagunas de la Reserva Biológica de Doñana (2018-2020). Geogaceta 2021, 70, 43-46.
- Acreman, M.; Casier, R.; Salathe, T. Evidence-based Risk Assessment of Ecological Damage due to Groundwater Abstraction; the Case of Doñana Natural Space, Spain. Wetlands 2022, 42:63.
- García-Novo, F.; Galindo, D.; García-Sanchez, J.A.; Guisande, C.; Jauregui, J.; López, T.; Mazuelos, N.; Muñoz, J.C.; Serrano, L.; Toja, J. Tipificación de los ecosistemas acuáticos sobre sustrato arenoso del Parque Nacional de Doñana. Actas del III Simposio del agua en Andalucía. Córdoba, 1991, Vol 1, pp. 165-176
- Díaz-Paniagua, C.; Fernández-Zamudio, R.; Serrano, L.; Florencio, M.; Gómez-Rodríguez, C.; Sousa, A.; Sánchez-Castillo, P.; García-Murillo, P.; Siljestrom, P. El Sistema de Lagunas Temporales de Doñana , una red de hábitats acuáticos singulares. Ed. Organismo Autónomo de Parques Nacionales. Miniterio de Agricultura, Alimentación y Medio Ambiente, Madrid, 2015. ISBN: 978-84-8014-880-1
- Fernández-Ayuso, A.; Rodríguez-Rodríguez, M.; Benavente, J.Assessment of the hydro-logical status of Doñana dune ponds: a natural world heritage site under threat. Sci. J 2018, 63 (15–16), 2048–2059.
- Fernández-Ayuso, A.; Aguilera, H.; Guardiola-Albert, C.; Rodríguez-Rodríguez, M.; Heredia,J.; Naranjo-Fernández, N. Unraveling the hydrological behavior of a coastalpond in Doñana National Park (Southwest Spain). Groundwater 2019, 57 (6), 895–906.
- Naranjo-Fernández, N.; Guardiola-Albert, C.; Aguilera, H.; Serrano-Hidalgo, C.; Rodríguez-Rodríguez, M.; Fernández-Ayuso, A., Ruiz-Bermudo, F., Montero-González, E.. Relevance of spatio-temporal rainfall variability regarding groundwater management challenges under global change: Case study in Doñana (SW Spain). Env. Res. Risk A. 2020, 34 (9), 1289–1311.
- Rodriguez-Rodriguez, M; Aguilera, H.; Guardiola-Albert, C.; Fernández-Ayuso, A. Climate influence vs. anthropogenic drivers in surface water-groundwater interactions in eight ponds of Doñana Natural Area (southern Spain). Wetlands 2021, 41(25).
Rev3-C8: The figures should be redrawn. Authors are recommended to refer to the instruction provided by the journal.
Rev3-C8Response: Thank you for the comment. We follow the instruction provided as we understand. Indeed we will be really happy if the reviewer could give us more details in order to solve this issue.
Rev3-C9: The method introduce is too brief.
Rev3-C9Response: Thanks. We added more information to further explain the use of the Morlet wavelet transform and added a new reference (172-175):
“Wavelet analysis utilizes the Morlet wavelet transform, which calculates the frequency structure across time, producing an optimal time-frequency resolution spectrogram [19-20, 23]. Morlet wavelet introduces a set of functions in the shape of small waves translated in time by and scaled in frequency from a simple generator function [24].”
Rev3-C10: The discussion should be made much deeper.
Rev3-C10Response: Thank you for the suggestion. Both Results and Discussion sections were inhanced and several changed have been introduced in lines (304-314):
“To further analysis of the variables water level – water temperature, cross wavelet analysis has been applied in the studied ponds. Several authors have pointed out the interest and importance of the application of this type of analysis in hydrological studies [37]. The cross wavelet can reveal the correlation degree between level and temperature variations at daily and sub-daily scales [24]. This way, it can be seen how temperature and water level are coherent in time and frequency. In figure 7, the cross-spectrograms of the Dulce -Santa Olalla hydrological system can be observed. Surface water level and temperature show strong daily joint periodicity, especially during summer and early autumn, related to evaporative cycles. However, on shallow groundwater, temperature and level exhibit consistent spectral wavelet correlation at a 0.5 days period throughout the entire series. ”
And lines 356-372:
“Olías-Alvarez [46] detected for the first time semi-diurnal tides in the DNP, in a piezometer at 6 km from the sea, with an amplitude of only 1 cm, much smaller than the amplitude measured by Fernández-Ayuso and Rodríguez-Rodríguez [47] in PSOL and PDUL, of 2 m, which seems to indicate that the tidal signal attenuates with respect to the distance to the Ocean in our study case.
Literature shows other examples of the detection of semi-diurnal tides in surface water. In Yucatan Peninsula, a karstified limestone area, some authors [48] found the tidal effect even in the water bodies more distant to the sea and proved that there was a clear connection between such water bodies and the regional aquifer, previously assumed to be hydrologically isolated from the regional aquifer. The tidal sign was also detected in shallow groundwater in ephemeral groundwater-fed lakes (turloughs)in Ireland [49], even at 8 km distance to the sea, in a karstic hydraulic system, much less transmissive than the aeolian sands in our study site.
Finally, the fact that in shallow groundwater temperature and groundwater level variables exhibit correlation at 0.5 days period can be attributed to dampened temperature variations caused by the force of tidal fluctuations driving heat transport by advection and dispersion, stated by other authors [38-40]. “
Reviewer 4 Report
General comments:
- There are many typographic and spelling errors throughout the manuscript. Please, revise the paper carefully. English language must be revised.
- The abstract needs to be rewritten by adding some quantitative results and the socio-econmic (added value) of this study.
- In the introduction section, the novelty of the work must be established in sections of introduction and discussion. It must be supported by recent research. And include suggestions that are attractive to other researchers in the world.
- The study area should be further detailed; especialy for hydrogeology characteristics.
- The discussion of the paper need to be enhanced; try to compare and to discuss your results with those by other researchers either in the same area or other regional sites?
- The conclusion must be rewritten with the main results.
Specific comments:
Line 47: delete double space between words in this line and check it throughout the text.
Line 48: replace "." by ","
Line 259: The reference style should be respected throughout the text (e.g. Borja et al.,).
More explaination about the noisy variations of water level and temperature in the S10 site.
Authors put only the hypothesis of the hydrogeological functioning of ZAH, SOL and DUL ponds. What about the other sites (S10)?
Author Response
REVIEWER 4
Rev4-C1: There are many typographic and spelling errors throughout the manuscript. Please, revise the paper carefully. English language must be revised.
Rev4-C1Response: Thank you. A thorough revision has been made and many typos and spelling errors have been solved in the new version of the MS.
Comment 2: The abstract needs to be rewritten by adding some quantitative results and the socio-economic (added value) of this study.
Rev4-C2Response: We acknowledge the reviewer’s suggestion. Indeed, the original Abstract lacks relevant information about the main findings of this paper. In the new version of the MS, a new paragraph has been added. In this paragraph, the new findings of this research have been included. For example, the 6-h interval signal, related to the Atlantic Ocean tides affecting the coastal aquifer and the 12-h interval signal, related to evaporation and evapotranspiration hydrological processes have also been included. The importance of this study in the understanding of the behavior of these ponds has also been mentioned in the new Abstract (Lines 16-34):
The Doñana National Park (DNP) is a protected area with water resources drastically diminish-ing due to unsustainable extraction of water for agricultural irrigation and human consumption of a nearby coastal city. In this study, we explore the potential of wavelet analysis applied to high temporal resolution groundwater and surface water time series of temporary coastal ponds in the DNP. Wavelet analysis was used to measure the frequency of changes in water levels and water temperature, both crucial to our understanding of complex hydrodynamic patterns. Results show that the temporary ponds are groundwater-dependent ecosystems of a through-flow type and are still connected to the sand dune aquifer, regardless their hydrological affection due to groundwater withdrawal. These ponds, even those most affected by pumping in nearby drills, are not perched over the saturated zone. This was proven by the evidence of a semi-diurnal (i.e., 6-hour) signal in the surface level time series of the shallow temporary ponds. This signal is, at the same time, related to the influence of the tides affecting the coastal sand dune aquifer. Finally, we detected other hydrological processes that affectthe ponds, such as evaporation and evapotranspiration, with a clear diurnal (12-hour) signal. The maintenance of the ecological values and services to the society of this emblematic wetland is currently at jeop-ardy due to the affection of the groundwater abstraction for irrigation. The results of this study contribute to the understanding of the behavior of these fragile ecosystems of DNP, and will also contribute to a sound integrated water resources management.” *Changes are underlined.
Comment 3: In the introduction section, the novelty of the work must be established in sections of introduction and discussion. It must be supported by recent research. And include suggestions that are attractive to other researchers in the world.
Rev4-C3Response: Thanks. In the introduction section a new paragraph has been added. The novelty and justification of this work has been explained in detail. In 2019, the authors published a paper in Groundwater (Fernández-Ayuso et al., 2019). This paper focused in the wavelet analysis of piezometric levels during a two-year period in Santa Olalla pond (Fig. 1, SOL pond) to enhance the hydrogeological conceptual model. This pond was not affected by anthropic disturbances. In this new paper the focus has been extended to four ponds, some of which show anthropic disturbances. Surface water time series (levels and temperatures) have been compared to piezometric water level/temperature time series (Lines 74-96):
“Wavelet Analysis used in the field of hydrology has been applied since the 1990s for multi-temporal scale analysis of hydrometeorological series, with increasing attention in the last decade [21-23]. Although most studies focus on the analysis of long climatic and hydrological time series to reveal inter-annual components [23-24], it has been shown [15] that this methodology is also helpful for shorter time series with a higher sampling frequency (e.g., hourly). The interpretation of the hydrological processes in a given water body (e.g., river, pond, playa-lake, aquifer, etc.) derived from a high temporal resolution but short (e.g., annual) time series will be different from a low temporal resolution but long time series (e.g., decadal). For this particular study, we have used a high-resolution (3-hour) yearlong time series to interpret the hydrological behavior of a series of ponds. The main reasons for this approach are: Firstly, the authors made a previous study in the same area but focused on the wavelet analysis of piezometric levels during a 2-year timespan [15] to enhance the hydrogeological conceptual model in one particular permanent pond, not affected by anthropic disturbances (Fig. 1, SOL pond). However, here, the focus has been extended to four ponds, some of them affected by anthropic disturbances [14], where surface and groundwater level and temperature time series have been compared. Secondly, for the first time, one of the ponds studied (DUL, see Fig.1) has been monitored with this high temporal resolution. Finally, the general hydrogeological behavior of the unconfined sand dune aquifer has been previously described, but not the interactions of this aquifer with the sand dune ponds included in this paper..”
Rev4-C4: The study area should be further detailed; especially for hydrogeology characteristics.
Rev4-C4Response: Thanks. Study area was enhanced and further details related with the hydrogeological evolution of the Guadalquivir estuary were included in the new version of the MS (lines 102-122) : “The DNP is a UNESCO World Heritage Site in southwestern Spain (37°N, 6°W) at the core of a huge wetland placed at the end of the Guadalquivir River delta. The Park has an extension of 54,252 ha, but it is connected to the hydrological catchment which is much bigger and covers an area of 260,000 ha. Originally, the area was a seasonal marsh system geologically formed by the siltation of a big estuary, approximately over the last 1,000 years. The estuary was slowly filled with silts and clays and transformed into a dynamic river, channel, and levee system. Then, a tidal marsh was formed and, finally, a fluvio-pluvial marsh filled the original estuary. The original marsh surface area has been transformed since 1920 and, at present times, only 1/5 of the surface remains in the original natural conditions. Climate is sub-humid and typically has rainfall of 550 mm per year, normally occurring between October - April. Some changes in temporal distribution in the rainfall days has been found by Naranjo-Fernandez et al. [22], high-lighting an increase of torrentiality. It rains the same amount of water but distributed in less days per year. The marsh is somewhat fed by a series of seasonal streams and, mainly, by direct precipitation. It is nowadays disconnected from the Guadalquivir River water course, although recent projects are taking course for its re-connection. The marshes are dry during summer due to high evapotranspiration rates. A series of active and stabilized aeolian dunes separates the marsh from the coast. These dunes host a series of ephemeral and temporary ponds that are dependent of groundwater discharge. Nevertheless, the structure and geology of the aquifer is complex.”
Rev4-C5: The discussion of the paper need to be enhanced; try to compare and to discuss your results with those by other researchers either in the same area or other regional sites?
Rev4-C5Response: Thanks. Results section was enhanced and several paragraphs have been included in the new version of the MS. Also, a new figure (Fig. 7) showing cross-spectrograms in Dulce - Santa Olalla hydrological system. It shows both Dulce and Santa Olalla (SOL) surface water cross correlograms and groundwater cross correlograms in “PDUL” (stands for Piezometer in Dulce pond) and “PSOLW”, stands for Piezometer in Santa Olalla pond (Western shore) (304-314):
“To further analysis of the variables water level – water temperature, cross wavelet analysis has been applied in the studied ponds. Several authors have pointed out the interest and importance of the application of this type of analysis in hydrological studies [37]. The cross wavelet can reveal the correlation degree between level and temperature variations at daily and sub-daily scales [24]. This way, it can be seen how temperature and water level are coherent in time and frequency. In figure 7, the cross-spectrograms of the Dulce -Santa Olalla hydrological system can be observed. Surface water level and temperature show strong daily joint periodicity, especially during summer and early autumn, related to evaporative cycles. However, on shallow groundwater, temperature and level exhibit consistent spectral wavelet correlation at a 0.5 days period throughout the entire series. ”
Rev4-C6: The conclusion must be rewritten with the main results.
Rev4-C6Response: Thanks. Conclusions were enhanced and some details about results have been included in the new version of the MS (lines 377-395).
“Through wavelet analysis and visual interpretation, we were able to provide evidence of a connection between surface water and regional groundwater systems in sand dune ponds of DNP, notwithstanding their hydrological affection due to groundwater withdrawal. The data series contained temperature and water level information in a 3-h time step of two shallow piezometers, one deep pumping well and four ponds, from May 2021 to May 2022. Such ponds are groundwater-dependent ecosystems of a through-flow type and therefore are not perched over the saturated zone as incorrectly stated by previous authors. Proven by the evidence of semi-diurnal tides affecting not only the groundwater measurements (sand dune regional aquifer) but also the surface water measurements (shallow ponds), and temperature, we were also able to detect other processes, such as evaporation or evapotranspiration. The results of this study have contributed to the improvement of the current knowledge about the hydrological behavior of shallow water bodies. It is worth to highlight the fact that it is the first time that the effect of the Doñana sand-dune aquifer on the water level oscillation of a shallow pond (c. 0,5 m depth) has been detected at an hourly interval. There were no other patterns to note in the water level and temperature data sets of the studied ecosystems. None less, we encourage water authorities to continue to monitor the situation so we may maintain this precious ecosystem.”
Specific comments:
Rev4-C7: Line 47: delete double space between words in this line and check it throughout the text.
Rev4-C7Response: Done
Rev4-C8: Line 48: replace "." by ","
Rev4-C8Response: Done
Rev4-C9: Line 259: The reference style should be respected throughout the text (e.g. Borja et al.,).
Rev4-C9Response: Done
Rev4-C10: More explaination about the noisy variations of water level and temperature in the S10 site.
Rev4-C10Response: In the new version of the MS, a new paragraph was included in order to explain the noisy variations in water level and temperature in the S10 (drill) site (lines 269-278): “As for the S10 (Fig. 5-A), in the months of July to September there are great noisy variations displayed due to summer groundwater abstractions in the coastal resort. Drops in the piezometric level of more than 20 m registered by the sensor installed inside the drill, almost instantaneously, and a subsequent and fast recovery of the level when the pumping stops, is the reason why wavelet results from July to September are numerical artifacts. As explained, due to the activation / deactivation of the pump in summer months, when the tourist arrive to the coastal resort, the oscillation of the water level is irregular. As for the temperature, the explanation is similar, because the drop in the water level leaves the sensor above the saturated zone inside the drill, so an abrupt change in temperature takes place (see Fig. 2 – F).”
Rev4-C11: Authors put only the hypothesis of the hydrogeological functioning of ZAH, SOL and DUL ponds. What about the other sites (S10)?
Rev4-C11Response: S10 is a drill, so for that reason is managed to pump water in summer months. In that sense, is difficult to extract sound scientific conclusions in this particular case.
Round 2
Reviewer 2 Report
I appreciate the authors for revising their manuscript, but the results appear incorrect. If the authors like to use one year of time series data, it is ok but the spectrograms and cross-spectrograms produced do not appear to be generated correctly.
Looks like the cross-spectrograms in Figure 7 are the same spectrograms in Figures 4 and 5. A cross-spectrograms is the cross-multiplication of the temperature spectrogram and water level spectrogram. Figure 7 does not appear to be correct and has also very poor quality.
Furthermore, how the spectrograms in Figures 4 and 5 are generated? The water level time series shown in Figure 3 have trends and jumps which can create low-frequency peaks in the spectrograms. Did the authors remove the trend and jumps before applying the spectrograms? Also, how did you treat the missing time series values/gaps?
Figure 7. What about the phase differences that are usually displayed by arrows on the cross-spectrograms? The arrows show how much the cycles lead/lag from each other. Please look at the references that I suggested for more details.
Figure 7. You also need to show 4 more cross-spectrograms for water level and rainfall time series. Including the phase arrows.
Furthermore, the overall presentation including English grammar requires significant work.
Thank you!
Author Response
REVIEWER 2
I appreciate the authors for revising their manuscript, but the results appear incorrect. If the authors like to use one year of time series data, it is ok but the spectrograms and cross-spectrograms produced do not appear to be generated correctly.
Question 1: Looks like the cross-spectrograms in Figure 7 are the same spectrograms in Figures 4 and 5. A cross-spectrograms is the cross-multiplication of the temperature spectrogram and water level spectrogram. Figure 7 does not appear to be correct and has also very poor quality.
Response 1: Thanks. The reviewer is right, we mixed-up the plots and we ended up placing the spectrograms shown in Figs 4 and 5 as the cross-wavelet transforms in Figure 7. We have now corrected it, and Fig 7 shows the actual cross-spectrograms between water level and temperature with improved quality.
Question 2: Furthermore, how the spectrograms in Figures 4 and 5 are generated? The water level time series shown in Figure 3 have trends and jumps which can create low-frequency peaks in the spectrograms. Did the authors remove the trend and jumps before applying the spectrograms? Also, how did you treat the missing time series values/gaps?
Response 2: To build the spectrograms in Figures 4 and 5, as well as the cross-wavelet power spectrums in Figure 7, the input time series is first detrended and then standardized. The WaveletComp package offers, as default, optional detrending using local polynomial regression (loess regression), and standardization is performed internally. Also, we set the upper period for wavelet analysis to 8 days to focus on the high-frequency components in the spectrograms.
We have added the following sentence in the Materials and methods section:
(205-213):
“Furthermore, before computing the wavelet power spectrum, the input time series are detrended using local polynomial regression and standardized. Both single wavelet transform and cross-wavelet transform analyses have been performed. The cross-wavelet transform is the analog of the covariance, and it allows comparing the frequency contents of two time series and drawing conclusions about the synchronicity of the series at specific periods and time ranges [20].”
(215-216):
“An upper period of 8 days was specified for wavelet decomposition to focus on the high-frequency components.”
On the other hand, there were very few missing values in the time series (i.e., < 1 %) randomly distributed that were filled through linear interpolation prior to wavelet analysis. The continuous high-resolution 3-hour measurement interval makes linear interpolation suitable to impute random gaps. Due to these facts, we hadn't mentioned the treatment of missing values in earlier versions of the manuscript so as to keep the document short. However, it is relevant information, as the reviewer points out, and we have added a sentence in the paper to mention it (205-207):
“Wavelet analysis requires complete records with no gaps, so the few randomly distributed missing values found in the time series (< 1 %) were imputed through linear interpolation.”
Question 3: Figure 7. What about the phase differences that are usually displayed by arrows on the cross-spectrograms? The arrows show how much the cycles lead/lag from each other. Please look at the references that I suggested for more details.
Response 3: “In Figure 7, the cross-spectrograms of Dulce, Santa Olalla, and Zahillo pond systems can be observed. Surface water level and temperature show strong daily joint periodicity, especially during summer and early autumn, related to evaporative cycles. Horizontal arrows pointing to the right indicate that the two series are in phase with vanishing phase differences (i.e., synchronized), especially for Dulce pond. These arrows are plotted only within white contour lines, indicating significance (with respect to the null hypothesis of white noise processes) at the 10% level. However, on shallow groundwater, level and temperature do not exhibit significant spectral wavelet correlation, although the highest cross-wavelet power in Santa Olalla belongs to a 0.5 days period throughout the entire series.”
Question 4: Figure 7. You also need to show 4 more cross-spectrograms for water level and rainfall time series. Including the phase arrows.
Response 4: We have not included cross-spectrograms for water level and rainfall because the latter is registered at daily time steps, not 3-hourly as the former. Therefore, the water level time series would have to be upscaled to daily intervals for cross-wavelet analysis, meaning that the highest frequency that could be analysed would be a 2-days period, preventing our study of high-frequency hydrological components of sub daily processes. Moreover, 1-year time series are too short to get any relevant information on hydroclimatic dynamics in semiarid environments that exhibit such irregular and variable precipitation patterns.
Question 5: Furthermore, the overall presentation including English grammar requires significant work.
Response 5: Thank you, we made a considerable effort to improve the overall presentation and the English grammar, considering the short time that we have to review it, which is not enough to send it to a English native speaker specialized in hydrogeology. We hope you notice the improvement.
Reviewer 3 Report
Authors did not addressed some comments raised in the first round, especially the hydrogeological setting which is significantly important.
Additionally, the figures (such as Fig.2) should be improved as the present version is too poor.
Author Response
REVIEWER 3
Question 1: Authors did not addressed some comments raised in the first round, especially the hydrogeological setting which is significantly important. Additionally, the figures (such as Fig.2) should be improved as the present version is too poor.
Response 1: As the reviewer mentioned, the hydrogeological setting was not addressed in the first revision of the MS. As for this revision, several paragraphs have been added in the new version, to address correctly the demands of reviewer 3: The first paragraph is related to the hydrological functioning and the transformation of the marshes:
“The marshes depend on rainfall and floods for their water supply and is classified as a pulse system. Water levels and extent of inundation are determined by direct rainfall and, more importantly by runoff arriving after rainfall in the river basin upstream, the Guadiamar River, an affluent of the Guadalquivir River. In the second half of the XXth century, the middle part of the Guadiamar basin and the marshes was anthropically disconnected by embankments. Consequently, the marshes became non-functional. Water was diverted to the Guadalquivir River.”
The second paragraph is broader and explains in detail the relation of all the Hydrogeological Units of the Almonte-Marismas aquifer. A new figure has also been added in the new version of the MS to better understand the characteristics of the aquifer. Two cross sections were included and explained in the text:
“…the sediments are mostly coastal, consisting of silts and sands. It is called the Deltaic Unit and goes from the Tinto River to the Guadiamar River (UD Unit in Fig. 0). These Quaternary sediments were sequentially covered by sands and gravels when the deltaic sedimentary environment was replaced by rivers, channels, and levees, typical of an alluvial environment. Such coarse sediments do not outcrop in Doñana, because they were as well gradually buried and replaced by different sedimentary rocks. It is called Alluvial Unit and it is depicted in CS-1 and in CS-2 of Fig. 0 (UA Unit, see figure caption). The Alluvial Unit was first covered by fine-grained sediments forming the so-called Marsh Unit: silts and clays that thicken from N (50 m) to S (70 m). See Fig. 1 for more details (UM Unit). Secondly, the most recent sedimentary environment is called the Aeolian Unit. This Aeolian Unit covers the Marsh Unit in the SE part of the aquifer and the Deltaic Unit in the SW part (see Fig. 1 and CS-1). This is a sandy dune belt, partly active nowadays that goes from the Tinto River estuary to the Guadalquivir River mouth. Finally, both deltaic, alluvial and marsh units are placed above the impervious substratum: marls and silts of the so-called Miocene and Pliocene Unit.
The Guadalquivir River Basin Authority (DHG) measures water levels in 300 piezometers across the aquifer. Some of the piezometric time series started in 1974. In many piezometers, groundwater levels are consistent with rainfall trends, but in other piezometers, levels were found to be declining.”
Figure 0. Hydrogeological sketch and structure of the Almonte-Marismas aquifer in southern Spain. The main Hydrogeological Units are as follows: L/MA stands for Miocene and Pliocene Unit (marls and silts). UD stands for Deltaic Unit (silts and sands). UM stands for Marsh Unit (clays and silts). UE stands for Aeolian Unit (fine sands). UA stands for Alluvial Unit. Below, a cross-section (CS) from W to E (CS 1) and from N to S (CS-2) can be seen. Modified from [29].
Finally, a short paragraph about the main use of the groundwater in the aquifer was included:
“Water is abstracted from the aquifer primarily for agriculture, particularly growing fruits, such as strawberries (now covering 6000 ha), cotton and rice.

Round 3
Reviewer 2 Report
Thank you for making the corrections. There are still a few things to be clear. Line 342. Do you mean at 90% confidence interval? Or is it 95%?
Figure 7. The arrows are too small and cannot be seen. Please enlarge them. Also, the quality of these figures are poor. Please improve them. A resolution of at least 300 dpi is needed.
Please carefully proofread the manuscript.
Author Response
Dear Reviewer 2,
First of all, we would like to thank you for all the comments that improve the original manuscript.
Taking attention over your questions:
Question 1: There are still a few things to be clear. Line 342. Do you mean at 90% confidence interval? Or is it 95%?
Response 1: We thank the comment. We realize that we are using the default confident interval of WaveletComp package in R. You are right, it is working 95% of confident interval. We change 10 for 5% in the line 347 of the new version of the manuscript.
Question 2: Figure 7. The arrows are too small and cannot be seen. Please enlarge them. Also, the quality of these figures are poor. Please improve them. A resolution of at least 300 dpi is needed.
Response 2: Thanks. In this case, the arrow length is not available to change but we improve the resolution of the figure. We hope the changes will improve the visualization of the Figure 7.
Question 3: Please carefully proofread the manuscript.
Response 3: During previous revisions, the authors of this article have made editorial improvements that have been saved with track changes. If any specific error is detected, we would appreciate it being indicated so that we can solve it as soon as possible.
Reviewer 3 Report
It can be considered for publication.
Author Response
Dear Reviewer,
We would like to thank for your comments and the opportunity to resubmit a revised copy of the manuscript.